# Pressure Analysis in Rigid and Flexible Real Arteriovenous Fistula with Thickness Variation In Vitro

**DOI:** 10.3390/jfb14060310

**Published:** 2023-06-02

**Authors:** Jonhattan Ferreira Rangel, Willyam Brito de Almeida Santos, Thércio Henrique de Carvalho Costa, Kleiber Lima de Bessa, José Daniel Diniz Melo

**Affiliations:** 1Materials Science and Engineering Post-Graduation, Department of Materials Science and Engineering, Federal University of Rio Grande do Norte—UFRN, Natal 59078-970, Brazil; 2Mechanical Post-Graduation, Department of Mechanical Engineering, Federal University of Rio Grande do Norte—UFRN, Natal 59078-970, Brazil

**Keywords:** arteriovenous fistula, pressure pulse, rigid, flexible

## Abstract

An arteriovenous fistula (AVF) is the access most recommended by several authors. However, its manufacture and use can cause several problems in the short, medium and long term. The study of fluid dynamics related to the structure of the AVF can provide information necessary for the reduction of these problems and a better quality of life for patients. The present study analyzed pressure variation in a rigid and flexible (thickness variation) model of AVFs manufactured based on patient data. A computed tomography was performed from which the geometry of the AVF was removed. This was treated and adapted to the pulsatile flow bench. Bench tests with simulation of systolic–diastolic pulse showed higher pressure peaks in the rigid AVF followed by the flexible model with 1 mm thickness. The inflection of the pressure values of the flexible AVF in relation to the rigid one was observed, being more expressive in the flexible AVF of 1 mm. The 1 mm flexible AVF presented an average pressure close to the physiological one and a smaller pressure drop, showing that this AVF model presents the best condition among the three to serve as a basis for the development of an AVF substitute.

## 1. Introduction

An arteriovenous fistula (AVF) is considered by most authors as the better vascular access for hemodialysis treatment [1,2,3,4]. However, this type of vascular access can be affected by problems related to the construction process and its prolonged use, which we can highlight: pseudoaneurysms, intimal hyperplasia, hand edema, among others [5,6,7,8,9].

The pressure short-circuit caused between the venous and arterial systems after the creation of the AVF has a direct influence on vascular hemodynamics. This change in flow ends up creating regions of recirculation in the flow (and consequent deposition of material and formation of thrombi), regions of high and low pressure, regions of stagnation that directly influence vascular remodeling and possible pathophysiological conditions in vascular access [6,10,11].

This remodeling is related to the increase or decrease in the thickness of the venous walls and the internal cross-section [8,10,12] and consequently a change in the deformation (or adaptation) capacity of these vessels according to the mechanical demands imposed by the increase in pressure in certain situations.

When surgery is performed, changes in flow conditions are expected, such as pressure-field variation, increased flow-rate and modification of the vascular structure due to dilation and increased wall thickness in the venous region. However, during the accompaniment of access maturation, it is possible to observe whether the AVF will succeed or fail. Several studies have sought to investigate the phenomena and parameters that influence the result of AVF maturation, highlighting pressure, flow rate and shear stress.

Several studies have aimed to analyze the AVF with a rigid wall, disregarding the influence of wall deformation on the flow of these accesses [9]. Variations in velocity, pressure and shear stress on vascular walls are parameters that directly influence the external remodeling of the vein (increase in vascular diameter). The high internal pressure also has a direct influence on the thickening of the venous wall [13] and the consequent increase in the stiffness of the vascular wall.

Some changes in AVFs are important for treatment. The most common and expected are related to the increase in the cross-section of the venous region (diameter) resulting from the pressure short-circuit and increased flow [14]. These structural changes are fundamental parameters for an effective treatment that guarantees a better flow-rate and less time in dialysis. Some studies relate the maturation of the AVF to the variation of the internal dimension of the vein in patients undergoing surgery to make the AVF [14].

The structural change in the venous and arterial region is also related to the variation in the thickness of its wall. Once the arteriovenous connection occurs, the veins that were previously subjected to a lower pressure tend to readjust to the more severe pressures causing hypertrophy of the venous wall. In experimental studies, variations in venous wall thickness in AVFs related to neointimal hyperplasia, increased extracellular matrix deposition and proliferation of smooth muscle cells were observed [14,15,16].

Under normal physiological conditions, the artery is subjected to higher pressures (120 mmHg) and has a greater wall thickness than the vein (80 mmHg). Based on these conditions, it is possible to infer that through the connection between the blood vessels and the increase in pressure in the venous region, the system that naturally does not have symmetrical thickness (artery and vein) tends to standardize its physical structure (thickness of the venous wall).

The in vitro study is an important tool to analyze these phenomena present in blood vessels and to develop solutions to pathophysiological problems. Among the criteria for generating an ideal in vitro model, the following stand out: the elastic mechanical response of the tissue to the pressure of the pulsatile flow, flow conditions mimetic to those present in the studied region and geometric characteristics similar to those of real blood vessels [17]. 

The present study investigated the pressure variation in the pulsatile flow in an AVF model made of rigid and flexible material (with wall thickness variation) seeking to mimic the structural changes resulting from the implementation of new physiological conditions in vascular access (short-circuit depression). The construction of the AVF was developed from data taken from a patient with this type of vascular access.

It is important to point out that this study deals with research and not a clinical analysis, since we sought to investigate the relationship between leakages in AVF models with different stiffness of the vascular walls, seeking to relate this parameter with the behavior of pressure along the access.

## 2. Materials and Methods

The study was divided into three stages: data acquisition and processing, AVF fabrication and experimental analysis.

### 2.1. Data Acquisition and Processing

#### 2.1.1. Data Acquisition—Computed Tomography

Medical data were collected using a computed tomography scan of the left upper limb of a patient with a brachiocephalic AVF. This procedure was performed at the Onofre Lopes University Hospital of the Federal University of Rio Grande do Norte (HUOL—UFRN).

After data collection, the InVesalius software was used for the three-dimensional reconstruction of living tissues and the selection of the mask that showed the region that made up the patient’s cardiovascular system. After the reconstruction and selection of tissues, the AVF was removed and subsequently cleaned. In Figure 1, it is possible to observe the three-dimensional reconstruction of the patient’s vascular system.

#### 2.1.2. Data Processing

The next phase of the work was the data processing. Initially, corrections were made in the virtual mesh generated by closing open elements resulting from the surface generation process. The next step in the process was the insertion of inlet and outlet ducts and pre-selected points for the investigation of pressure (two points in the arterial region, one point in the anastomosis and two points in the venous region—pseudoaneurysms), as shown in Figure 2.

The flow in the study developed from the proximal artery to the proximal vein without the influence of flow in the distal artery of the access, so the AVF had only one inlet and one outlet, not mimicking the consequence of flow through other parts of the access (distal artery) as the ischemic steal syndrome.

### 2.2. AVF Manufacturing

3D printing was used to manufacture both AVF models. In the rigid model, the printing of the virtual model of the AVF was performed in compact material. For the flexible models, a mold and internal volumes with dimensional variation of the venous region were developed.

#### 2.2.1. Flexible AVF

Flexible AVFs were manufactured using an injection mold and an internal volume (male), both made by 3D printing. Two flexible AVFs were developed with different venous thicknesses. The first AVF manufactured was 2 mm thick across its entire length. To make this model of an AVF, a mold was designed based on the geometry taken from the patient with an additional thickness of 2 mm and an internal volume compatible with the original dimension of the AVF. The mold and internal volume are shown in Figure 3.

The AVF with a reduced-thickness venous region (1 mm thick) was made with the same mold mentioned above, making adjustments only in the dimension of the venous region of the internal volume. A 0.5 mm shell was added around the entire venous extension of the anastomosis until the access exit, reducing the space between the mold and the element centered in this region.

During the entire manufacturing process, the created AVFs were inspected, looking for possible flaws that would make it impossible to use them in tests on experimental workbenches. After curing the silicone, the internal volume was removed using acetone at 98% concentration (CH_3_COCH_3_) (ATRIOM—Chemical products LTDA).

#### 2.2.2. Rigid AVF

Based on the geometry developed in the previous steps, a 3D printing of the rigid model of the AVF was performed. Among the parameters used, we can highlight: the material, the thermoplastic polymer ABS (acrylonitrile butadiene styrene), and the 100% filling, ensuring rigidity and preventing leaks during the tests.

### 2.3. Experimental Analysis

After the fabrication of the AVF models, tests on an experimental workbench to simulate the systolic–diastolic pulse were performed (Figure 4). The workbench was subdivided into: control, test, sensors and data sections. Arrows indicate the flow of experimental steps between sections of the experimental bench.

The fluid used during the experiments was water at 25 °C (specific mass of 997.05 kg/m³ and dynamic viscosity of 1.1E-3 Pa.s). As the work sought to analyze the influence of the variation in AVF stiffness on the pulsatile flow, there was no need for a fluid with a temperature and viscosity similar to blood, and these characteristics could be used in a subsequent analysis.

The pressure pulse implemented was based on 4 distinct mathematical functions (see Table 1) that together formed the physiological pulse 120 × 80 mmHg (approximately 16 × 10.7 kPa)—Figure 5.

The results presented are in the International System of Measurements (S.I.), so the pressure values are presented in kPa (kiloPascal) on the y axis.

### 2.4. Analysis Equations

As the flow in the flexible AVFs causes deformation in the system, we can analyze the pressure from the perspective of the variation of the thickness and internal area of the system. First, as the deformations due to yielding occur in the elastic region of the material, we have:(1)σ=E·ε
where “σ” is the normal stress of the system (Pa), “E” is the modulus of elasticity of the material (Pa) and “ε” is the deformation suffered according to the stress imposed on the system. Based on Laplace’s law, for a tube the circumferential strain can be described as:(2)$$\sigma =\frac{P\cdot r}{h}$$
where “P” is the system pressure, “r” is the radius and “*h*“ is the vessel wall thickness. Relating Equations (1) and (2) we have:(3)$$\frac{P\cdot r}{h}=E\cdot \varepsilon$$

It is possible through Equation (3) to observe that the pressure in the system varies according to the radius (area variation) and thickness for the same material and the same applied stress.

## 3. Results

### 3.1. AVF Manufacturing—Results

The first results obtained were the molds, the internal volumes, the rigid AVF and the flexible AVF models.

The rigid AVF model (Figure 6) presented leakage problems during the preliminary tests, which were corrected through the process of external painting with epoxy resin, not influencing the geometry.

The fabrication of the flexible AVF started with the fabrication of the mold. The molds developed had problems with centering, which were corrected by inserting a central plane in the entire geometry of the AVF, coinciding with the center line of the entire AVF, in addition to using the accesses themselves as points for centering. The manufactured mold can be seen in Figure 7.

During the manufacture of these FAVs, some bubbles were observed inside; this problem was corrected by injecting the silicone under pressure into the mold. Figure 8 shows a fabricated flexible AVF model.

### 3.2. Pressure Analysis—Pulsatile Flow

Pressure values at points (1), (2), (3), (4) and (5) were collected and analyzed in the three AVF models during the experiments with pulsatile flow. The pressure pulses at the points described above in the three AVF models are shown below—Figure 9.

The values of the resulting pressure at the specific point of the AVF as a function of normalized time (axis x) are observed on the y axis. These values result from the variation in the geometry and thickness of the AVF and its implications for flow.

In Figure 9 it is possible to observe the magnitude of the pressure pulses and their maximum values, in addition to the inflection regions of the flexible AVF pulses as a function of the rigid ones. The inflection between the flexible 1 mm AVF model and the rigid AVF occurred in a normalized mean time of 0.35 and between the flexible 2 mm model of 0.805.

The peak pressure values observed at pressure points (1), (2), (3), (4) and (5) and the outlet are shown in Table 2. Higher pressure values were observed in rigid AVFs at all points, except for the 1 mm flexible AVF output. Between the two flexible AVF models, the highest pressure values were observed in the model with 1 mm wall thickness.

Points (4) and (5) showed high pressure values when compared to the exit point. These points presented these values due to their location in the pseudoaneurysms resulting from the punctures.

## 4. Discussion

Through the results obtained, it was possible to observe the influence of the variation of the thickness and diameter of the venous region on the systemic pressure. As expected, the highest pressure values were observed in the rigid AVF due to the non-damping of the flow due to the deformation of the vascular wall. Vascular wall hypertrophy and increased stiffness associated with physiological, metabolic and physical factors cause a general increase in pressure [11].

The highest pressure values after those presented by rigid AVFs were from the flexible AVF model with 1 mm of vascular wall thickness. This thickness can be considered as the initial structure of the venous region immediately prior to the pressure short-circuit caused by the construction of the AVF. Based on this access characteristic and through Equation (3), it can be observed that from Laplace the pressure is directly proportional to the vessel thickness variation and inversely proportional to the radius (cross-section). We can thus observe that even with the decrease in thickness and increase in the area of the vascular region, higher pressures were obtained in this model than in the 2 mm AVF model. This value of pressure in the AVF may be more linked to the deformation of the system than to the variation in the thickness of the access, since the AVF with the same input load (pressure pulse) and greater wall thickness tends to dissipate a greater amount of energy through the walls than the one with less thickness and greater area.

Another justification for this is through the analysis of the variation of energy and amount of mass of the flow, where the increase in the area influences the decrease in speed and consequent increase in local pressure, as evidenced in the venous region of the access.

Evaluating the variation of the pressure value under the physiological aspect, it is common knowledge that the veins are vessels that, among other functions, are responsible for storing part of the blood flow to use in the regulation of this when necessary. As the variation of energy in the blood flow can be related to the relationship between pressure and volume, once there is a variation in the internal volume of these vessels, a variation in their local pressure must therefore occur. The yield strength is inversely proportional to the square of the radius.

The evidence showed that the non-symmetry of the AVF causes an increase in pressure in the venous region, which is not physiologically interesting for access. Therefore, when the thickness of the vein increases and the AVF becomes symmetrical, the value of venous pressure decreases, reducing pathological problems related to the uncontrolled thickening of certain regions (neointimal hyperplasia) of the vascular access and the circumferential tension in the access [5].

It can then be inferred that the 1 mm thick vein seeks, through the thickening of its wall, a readjustment to the new flow conditions, trying to approach the structural characteristics of the arterial region. Therefore, when it becomes symmetrical (constant thickness throughout the AVF) the access distributes the deformations better and consequently the flow energies tend to become uniform, reducing the stresses on the access walls.

The highest pressure value was observed at point (1) of the 1 mm flexible AVF. This is due to its location close to the access entrance. It is important to note that pressure values cannot be interpreted purely on the basis of their magnitude; their location and cross-sectional size must also be taken into account.

Point (2), located in a curved region of the arterial system, presented relatively high pressure values when considering its location in the system. This fact occurred due to its location in a curved region and the way the sensor captured part of the dynamic pressure, presenting high values. In the other regions the pressure taps were collecting static pressure. These pressure values show that this point is located in a place that suffers continuous flow shock and is, therefore, a region susceptible to pathophysiological problems.

Some longitudinal studies in humans associated venous remodeling with a decrease in shear stress on the wall and consequent reduction in the formation of neointimal hyperplasia in the access [19,20,21,22,23] and increase in the venous wall related to the attempt to decrease the pressure intraluminal [11]. That is, the remodeling of the access is an intrinsic parameter in controlling severe changes in the flow, both its dilation and the thickening of its venous wall, but the control of the latter (thickness of the venous wall) is fundamental to guaranteeing better conditions of flow, as evidenced in the results of the work.

Point (3) located in the system anastomosis presented high pressures related to its cross-sectional area. This region is one of the main sites for studies related to maturation and possible pathophysiological conditions related to changes in flow [22] Due to the chaotic flow, the pressure values can vary according to the place of the anastomosis where they are collected, presenting regions of low pressure (regions of recirculation) and regions of high pressure (stagnation points). The point chosen for the pressure collection is in the upper and central plane of this region. It is believed that the values are those of a recirculation zone.

Points (4) and (5) presented the highest relative values of system pressure when compared to the pressure at the access outlet. This fact occurred due to the locations of these points in regions of greater cross-sectional area, the pseudoaneurysms, resulting from the puncture for the treatment.

Recent studies show that hemodynamics is one of the main factors regarding the formation of aneurysms [17]. Therefore, geometry (shape) together with hemodynamics are fundamental study elements for the short- and long-term analysis of pseudoaneurysms generated in the AVF, how they behave and the influence they have on the overall flow of the access vascular.

The inflection points in the graphs of Figure 9 are located where the pressure values in the flexible AVF assume greater values than in the rigid AVF. These phenomena occur at two different times for different AVFs: in the normalized mean time of 0.805 for flexible 2 mm AVF and 0.35 for flexible 1mm AVF. This inflection occurs due to the return of part of the pressure energy responsible for the deformation of the FAV walls to flow. Therefore, in vessels with higher or lower deformation capacity these values will be higher or lower [22]. Thus, as the 1 mm thick, flexible AVF has a greater cross-sectional area in the venous region and is less thick, the values of energy returned by the vessel walls to the flow were higher when compared to the 2 mm flexible AVF. This greater amount of energy stored by the walls of the 1 mm AVF also explains the greater speed in its return to the flow.

The values for the average pressure drop between Point (1) and the exit of the access from 9.85 kPa to 6.84 kPa to 7.59 kPa in the rigid AVF, flexible AVF—1 mm and flexible AVF—2 mm, respectively. Through these values, it can be inferred that the flexible AVF with less thickness in the venous region caused a lower pressure drop in the system; therefore, the energy provided at the entrance of this access model was maintained for a longer time, reducing the pump effort (physiologically—heart) for the same required flow.

Among the characteristics previously mentioned for an ideal in vivo study, we can highlight another: presence of the cells responsible for the response to the mechanical stimulus, such as smooth muscle cells and endothelial cells [17]. However, as this work sought to investigate the flow response to the change in geometry and the elastic response of a synthetic material, the presence of these elements did not significantly influence the result.

This relationship of inflection and energetic interaction between fluid and structure is a parameter that has been little studied, but is extremely necessary for the development of new materials and their use as vascular grafts, which, if poorly dimensioned, can overload systems that already have pre-existing problems.

It is important to point out that this study takes into account the purely mechanical parameters of the flow, not relating the physiological phenomena that act in the regulation of blood flow.

## 5. Conclusions

Through the results obtained, the following values of peak pressure were observed in a decreasing way: rigid AVF, flexible AVF of 1 mm and flexible AVF of 2 mm. The higher pressure value in the rigid AVF was related to the non-amortization of the flow-pressure pulse, which causes energy losses to be related to friction and localized losses (shocks and changes in the flow direction).

The pressure values in the flexible AVF showed that the decrease in the venous thickness of the flexible AVF associated with the increase in internal area caused a local increase in the pressure, which was related to the energy absorbed, dissipated and returned to the flow, the variation of the internal volume and modification of the flow energies related to the variation of the internal cross-sectional area.

Another characteristic influenced by the change in the thickness of the venous region was the inflection in the value of the pressure pulses, which was faster in the AVF with 1 mm of venous thickness than in the one with 2 mm of venous thickness, a parameter also related to the difference in pressure between these two models of flexible AVFs. The increase in the cross-section ensured a decrease in the total energy-loss of the system.

Analyzing the results, we can conclude that the recommendation would be the internal remodeling of the venous region of the AVF (increase in the cross-section followed by maintaining the thickness of this region). These two linked parameters caused an increase in pressure in the venous system and helped in the puncture process necessary for hemodialysis treatment, in addition to providing better flow-rates, less debilitating the patient.

Ways to solve AVF problems are related to the variation of the anastomosis angle, the area of connection between artery and vein and control of drainage conditions, among others. Aiming at material engineering, it is emphasized that the use of materials in the AVF, or grafts, must guarantee mechanical conditions close to the physiologies of, and biocompatibility with, the human body. The 1 mm flexible AVF showed a pressure close to the physiological one with a lower pressure drop between inlet and outlet, therefore, being able to provide better conditions for the patient’s treatment.

## Figures and Tables

**Figure 1 jfb-14-00310-f001:**
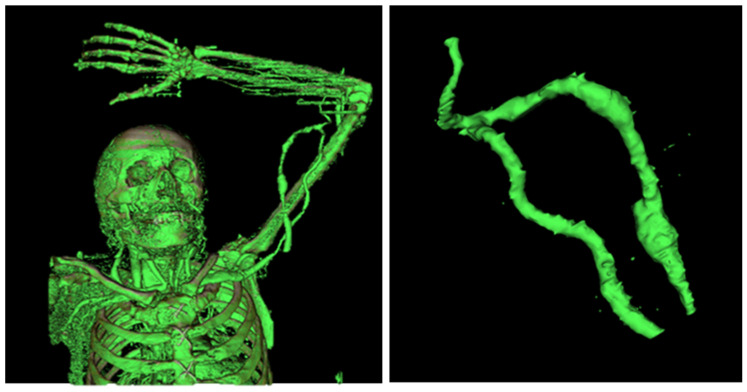
Three-dimensional reconstruction and AVF—InVesalius.

**Figure 2 jfb-14-00310-f002:**
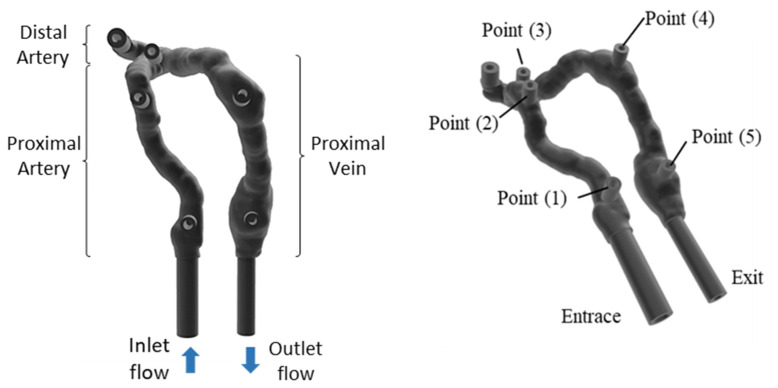
AVF geometry—virtual model.

**Figure 3 jfb-14-00310-f003:**
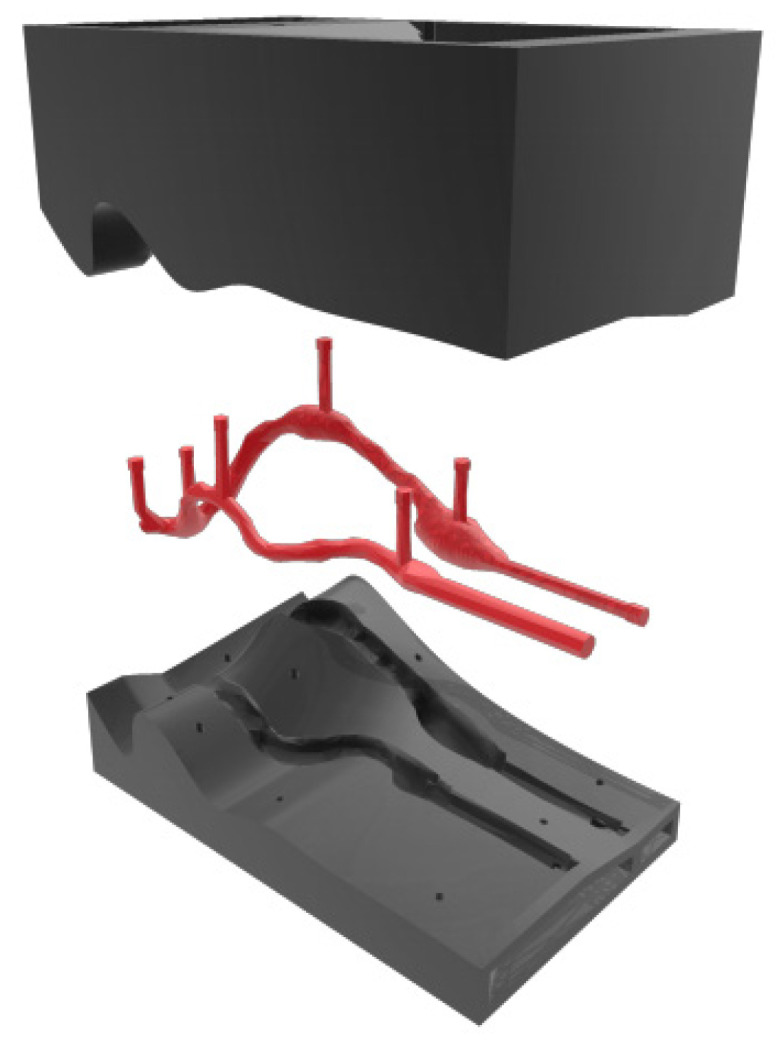
Mold and internal volume—virtual model.

**Figure 4 jfb-14-00310-f004:**
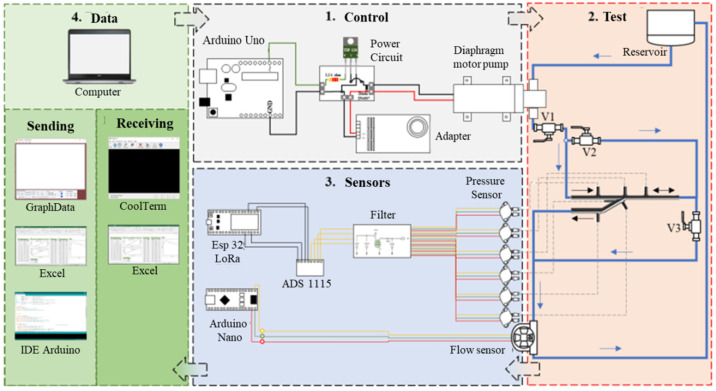
Experimental workbench—schematic drawing—adapted [18].

**Figure 5 jfb-14-00310-f005:**
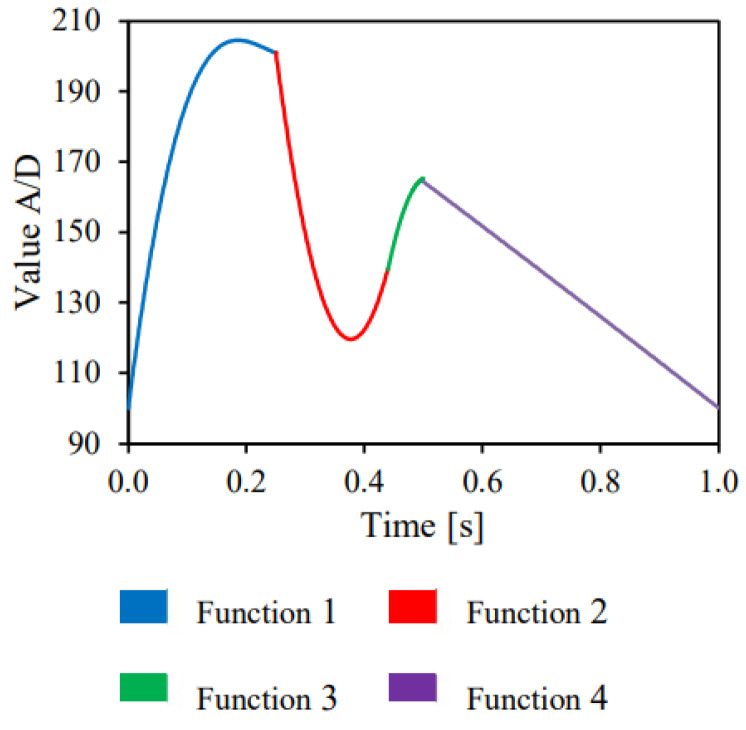
Pressure pulse equations—adapted [18].

**Figure 6 jfb-14-00310-f006:**
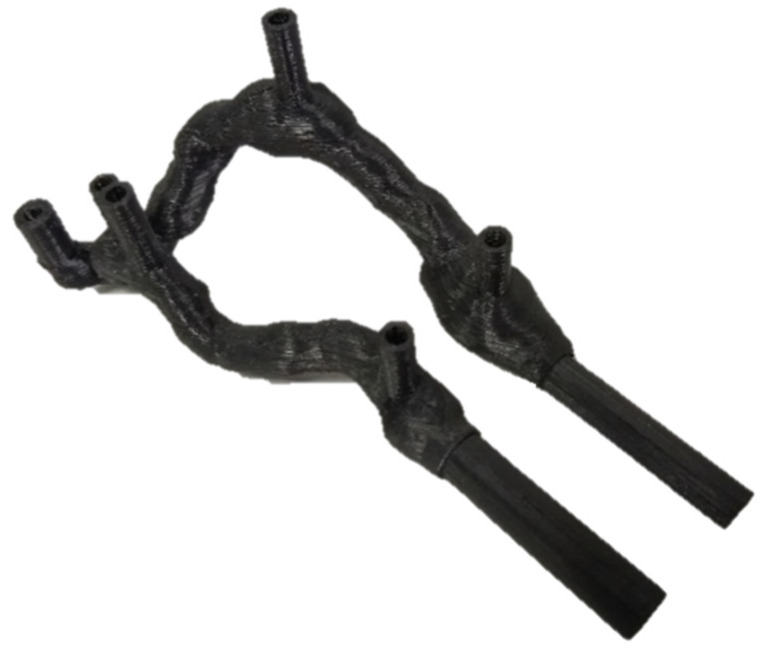
Rigid AVF—result.

**Figure 7 jfb-14-00310-f007:**
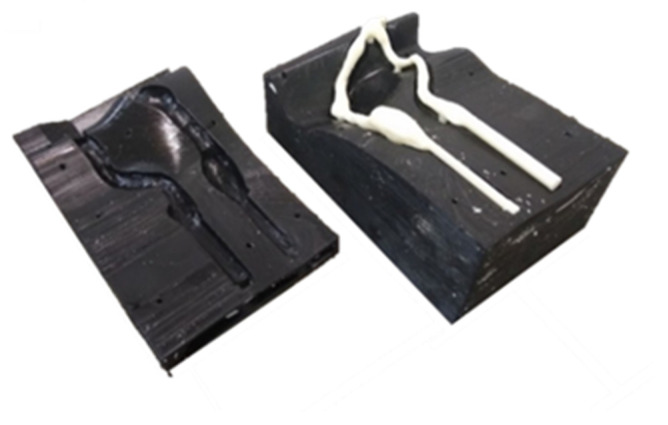
Mold and internal volume.

**Figure 8 jfb-14-00310-f008:**
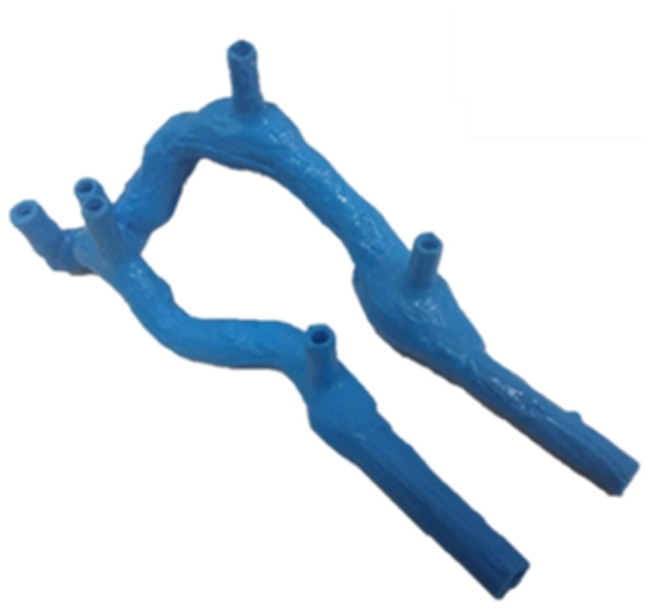
Flexible AVF.

**Figure 9 jfb-14-00310-f009:**
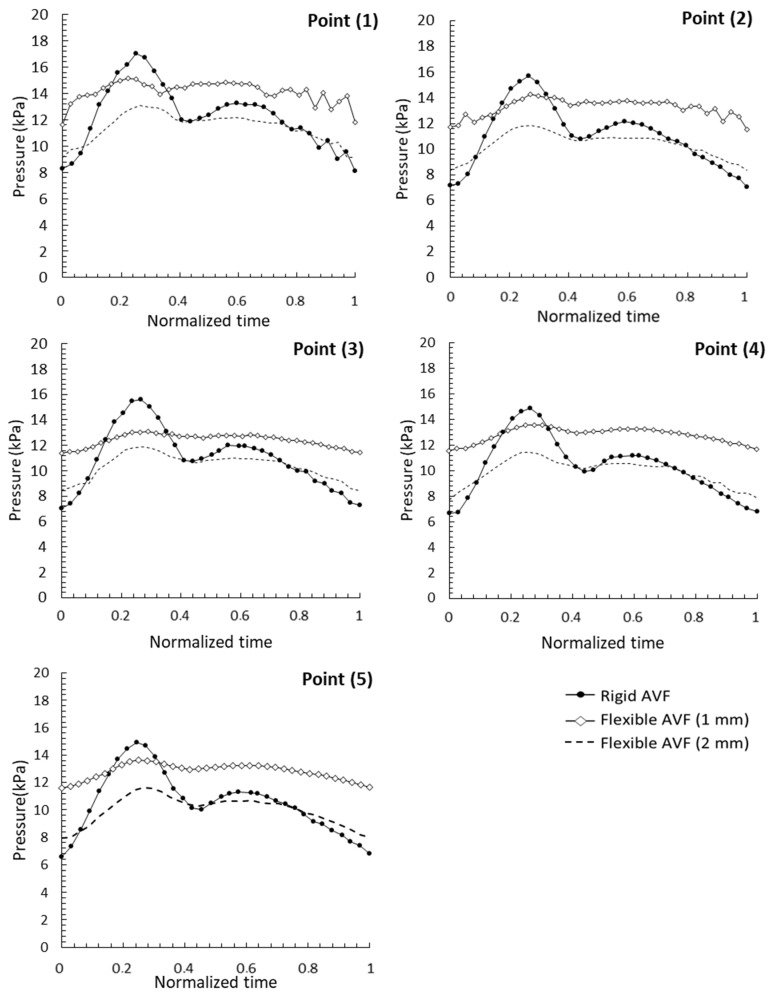
Pressure pulse—points (1), (2), (3), (4) and (5).

**Table 1 jfb-14-00310-t001:** Characteristic functions of the pressure pulse.

Function 1	y = 11.138x^4^ + 1607.1)x^3^ − 4774.4x^2^ + 1322.5x + 100
Function 2	y = 5060.4x^2^ − 3814.3x + 838.37
Function 3	y = −5833.3x^2^ + 5915x − 1333.9
Function 4	y = −4.1436x^2^ − 122.24x + 226.49

**Table 2 jfb-14-00310-t002:** Peak pressure (kPa).

Peak Pressure (kPa)						
	Point (1)	Point (2)	Point (3)	Point (4)	Point (5)	Exit
AVF rigid	17.02	15.72	15.59	14.87	14.90	7.17
AVF flexible (1 mm)	15.16	14.25	13.03	13.56	13.64	8.32
AVF flexible (2 mm)	13.11	11.87	11.88	11.42	11.63	5.52

## Data Availability

All data supporting the results of this manuscript has been included in this manuscript along. Supplementary data can be provided from the corresponding authors on request.

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
