# Peer review of "Pressure Analysis in Rigid and Flexible Real Arteriovenous Fistula with Thickness Variation In Vitro"

_jfb, 2023, doi:10.3390/jfb14060310_

Round 1
Reviewer 1 Report
The present study analyzed 15 pressure variation in a rigid and flexible (thickness variation) model of AVF manufactured based 16 on patient data. A computed tomography was performed from which the geometry of the AVF was 17 removed, which was treated and adapted to the pulsatile flow bench. This study showed innovation, and completed well. Therefore, minor revision is recommended to improve the quality of the figures and text organization.
Thank you for your reminder. Like some of the issues of the figures should be enhanced.
(1) The main question addressed by the research is Arteriovenous fistula (AVF) has some issues need to be refined in clinical use. The author harvested some data to stimulate the process clinically, and try to figure out the solution. However, the conclusion failed to give the support to refine the AVF.
(2) This study seems like a clinical report instead of a study. The conclusion is lacking in the abstract.
(3) The manuscript needs to be re-organized to be a basic research not a clinical report.
(4) The figures need refined. Like figure 9, the author should make a layout to join the diagrams with the pressure pulse images. Also, the necessary and important information should be provided to support the conclusion.
(5) The references need to be enriched, no less than 20 references will enhance the manuscript.
Author Response
Dear reviewers,
We thank the excellent reviewers for their opinions and scores, aiming to improve the work entitled “Pressure analysis in rigid and flexible real arteriovenous fistula with thickness variation in vitro.” It is important to emphasize that we agree with the opinions and suggestions which were added to the manuscript to improve its scientific character, both those related to the way the work was formatted and those of a theoretical nature.
Below are the answers to the questions asked by the reviewers and the corrections made to the manuscript.
Revisor #1
(1) The main question addressed by the research is Arteriovenous fistula (AVF) has some issues need to be refined in clinical use. The author harvested some data to stimulate the process clinically, and try to figure out the solution. However, the conclusion failed to give the support to refine the AVF.
ANSWER: Dear reviewer, we would first like to thank you for your scores and emphasize that a final opinion was added to the conclusion section on which type of arteriovenous fistula (AVF) would be most suitable to fulfill its functions and characteristics, aiming at the mechanical and physiological well-being of the AVF and patient's clinic.
(2) This study seems like a clinical report instead of a study. The conclusion is lacking in the abstract.
ANSWER: Dear reviewer, we are grateful for your score. In the introduction of the work, we added punctuation regarding the scientific research nature of the work, seeking to clarify the nature of the work to the readers, not being related to a clinical report, since we sought to investigate the influence of the deformation of the vascular wall on the blood pressure. AVF The AVF was manufactured using data from a real patient; however, the analysis was performed entirely on a bench, and the fabrication of the rigid and flexible models was mechanical.
(3) The manuscript needs to be re-organized to be a basic research not a clinical report.
ANSWER: Dear Reviewer, we have organized parts of the work to improve and exhaust its characteristics as a scientific text and research. Thanks for the improvement suggestions.
(4) The figures need refined. Like figure 9, the author should make a layout to join the diagrams with the pressure pulse images. Also, the necessary and important information should be provided to support the conclusion.
ANSWER: Dear reviewer, as requested, we adjusted image 9 and improved the quality of the other images in the manuscript, in addition to adding some more information about the results and their interpretative analysis, such as the mean pressure drop, and the physiological relationship between the pulses and the conclusion end of work.
(5) The references need to be enriched, no less than 20 references will enhance the manuscript.
ANSWER: Dear reviewer, thank you for your concern and suggestion for enriching the work. New references were added to the manuscript as requested.

Reviewer 2 Report
The authors use rigid and flexible models of AVF based on patient datato analyze pressure changes .
The results provide very interesting and important information for the reader, however the following points are required to be revised.
Major points.
The fluid used in the experiment is water at 25 ℃, but blood is 36-37 ℃ and has viscosity, so the data obtained in this study may not match. This point should be discussed.
The authors used physiological pulse (120 x 80 mmHg), but the relationship to the vertical axis of the experimental results with blood pressure is unclear.
Author Response
Dear reviewers,
We thank the excellent reviewers for their opinions and scores, aiming to improve the work entitled “Pressure analysis in rigid and flexible real arteriovenous fistula with thickness variation in vitro.” It is important to emphasize that we agree with the opinions and suggestions which were added to the manuscript to improve its scientific character, both those related to the way the work was formatted and those of a theoretical nature.
Below are the answers to the questions asked by the reviewers and the corrections made to the manuscript.
Revisor #2
(1) The fluid used in the experiment is water at 25 ℃, but blood is 36-37 ℃ and has viscosity, so the data obtained in this study may not match. This point should be discussed.
ANSWER: Dear reviewer, thank you for the score regarding the characteristics of the experimental fluid. As the work seeks to demonstrate the relationship between the pressure difference in rigid and flexible AVF, it was decided to use water in the tests since the properties of the fluid would be constant in all experiments. Future works will be developed to investigate the influence of viscosity and temperature. In the literature, the use of water as a working fluid is evidenced when the authors seek to investigate only the influence of geometry on the flow. I inserted a point in the methodology section of the manuscript showing the reason for using water as a working fluid.
(2) The authors used physiological pulse (120 x 80 mmHg), but the relationship to the vertical axis of the experimental results with blood pressure is unclear.
ANSWER: Dear reviewer, we are grateful for presenting this question. The value implemented in the fistula was approximately 120 x 80 mmHg (16 x 10.7 kPa). The results shown on the y-axis of the graphs are in the international measurement system (SI) as it is a survey published in an international journal. Therefore, we chose to present it more didactic; however, based on your review, we added a sentence in the methodology justifying the use of IS in the results presented.